# A genetic basis for molecular asymmetry at vertebrate electrical synapses

Adam C Miller[1]*[†][‡], Alex C Whitebirch[1][†], Arish N Shah[1], Kurt C Marsden[2], Michael Granato[2], John O'Brien[3], Cecilia B Moens[1]

[1]Division of Basic Sciences, Fred Hutchinson Cancer Research Center, Seattle, United States; [2]Department of Cell and Developmental Biology, University of Pennsylvania Perelman School of Medicine, Philadelphia, United States; [3]Department of Ophthalmology and Visual Science, McGovern Medical School, University of Texas Health Sciences Center at Houston, Houston, United States

**Abstract** Neural network function is based upon the patterns and types of connections made between neurons. Neuronal synapses are adhesions specialized for communication and they come in two types, chemical and electrical. Communication at chemical synapses occurs via neurotransmitter release whereas electrical synapses utilize gap junctions for direct ionic and metabolic coupling. Electrical synapses are often viewed as symmetrical structures, with the same components making both sides of the gap junction. By contrast, we show that a broad set of electrical synapses in zebrafish, *Danio rerio*, require two gap-junction-forming Connexins for formation and function. We find that one Connexin functions presynaptically while the other functions postsynaptically in forming the channels. We also show that these synapses are required for the speed and coordination of escape responses. Our data identify a genetic basis for molecular asymmetry at vertebrate electrical synapses and show they are required for appropriate behavioral performance.

*For correspondence: acmiller@uoregon.edu

[†]These authors contributed equally to this work

Present address: [‡] Institute of Neuroscience, University of Oregon, Eugene, United States

Competing interests: The authors declare that no competing interests exist.

## Introduction

Brain function is derived from the patterns of connectivity and the types of synapses made between neurons within a circuit. Synapses come in two main types, chemical and electrical, that together dynamically define neural circuit function across all life stages and animal phyla (*Connors and Long, 2004*; *Meier and Dermietzel, 2006*; *Pereda, 2014*). Chemical synapses are broadly used within the nervous system and are overtly asymmetric at the molecular and functional level, with communication between neurons achieved via presynaptic neurotransmitter release and postsynaptic neurotransmitter reception. Electrical synapses are best known for their roles in early nervous system development, however they are found throughout the brain from inception to adulthood and contribute to function from sensation to central processing to motor output (*Hormuzdi et al., 2004*). Electrical synapses are generally viewed as molecularly and functionally symmetric, allowing for very fast bidirectional ionic and metabolic neuronal communication (*Connors and Long, 2004*). This direct route of communication is achieved by neuronal gap junctions (GJs) that are formed from plaques of tens to thousands of channels between the neurons (*Raviola and Gilula, 1973*; *Hormuzdi et al., 2004*). While often viewed as simple channels, work in invertebrates has found that unique GJ-forming proteins can be contributed asymmetrically from each side of the synapse (*Phelan et al., 2008*; *Starich et al., 2009*). Moreover, the molecular asymmetry can create functional asymmetry (rectification) in ionic flow through the GJ (*Phelan et al., 2008*). In vertebrate nervous systems electron microscopy and electrophysiology suggests that electrical synapse structure and function can also be asymmetric (*Brightman and Reese, 1969*; *Rash et al., 2013*). However, the

genetic basis for such molecular asymmetry at vertebrate electrical synapses has never been identified.

Neuronal GJs are intercellular channels that establish cytoplasmic continuity between two cells allowing ions and molecules less than ~1 kilodalton to traverse between cells. Each channel is composed of two hemi-channels, one contributed by each neuron, that interact via their extracellular domains to form the channel (*Figure 1A*) (*Connors and Long, 2004*). Vertebrate GJ-forming proteins are the Connexins (Cx) while invertebrates use an evolutionarily distinct but functionally equivalent set of Innexins (*Phelan, 2005*). In mammals there are 21 genes encoding Cx proteins of which five are known to be expressed in neurons and to form electrical synapses (*Connors and Long, 2004*; *Söhl et al., 2005*). A partial genome duplication that occurred in teleost fish has increased the number of Cx-encoding genes, and the ensuing evolutionary trajectories saw some genes being duplicated and others lost (*Eastman et al., 2006*; *Cruciani and Mikalsen, 2007*; *Zoidl et al., 2008*). This diversity of Cxs encoded in the genome results in cases where individual cells or apposed cells express multiple different GJ-forming proteins providing the opportunity for heteromeric hemichannels made from multiple Cxs or heterotypic channels made by pairing hemichannels between cells with different Cxs (*Palacios-Prado et al., 2014a*). In vertebrates, heterotypic GJs between cells in non-neuronal tissue are common (*Goodenough and Paul, 2009*), however it has been notoriously difficult to identify heterotypic Cx configurations at electrical synapses in neurons.

Here we identify two Cxs used asymmetrically for the formation and function of electrical synapses in zebrafish. We focus on the Mauthner neural circuit, which mediates a fast escape behavior, since it has uniquely identifiable pre- and postsynaptic neurons making stereotyped electrical synapses that contribute to the escape response. We find that two Cx-encoding genes, *gjd1a/cx34.1* and *gjd2a/cx35.5*, are required for synapse assembly and function. Both of these Cx proteins are localized to synapses throughout the Mauthner circuit, as well as other synapses within the hindbrain and spinal cord, and each requires the other for recruitment to the synapse. Using chimeric analysis we demonstrate a dramatic asymmetry at electrical synapses in the circuit: *gjd1a/cx34.1* is necessary and sufficient postsynaptically while *gjd2a/cx35.5* is necessary and sufficient presynaptically for synapse formation. Using high throughput behavioral analysis we find that the electrical synapses contribute to the speed and coordination of the Mauthner-induced escape response. Together our data support a model wherein molecularly asymmetric electrical synapses act in concert with chemical synapses to impart performance onto the neural network output.

## Results

### The *disconnect3* mutation disrupts *gjd1a/cx34.1* and electrical synapse formation

To investigate genes required for electrical synapse formation in vivo we used the Mauthner (M) neural circuit of larval zebrafish (*Figure 1B*). The Mauthner circuit coordinates an escape response to a variety of threatening stimuli and produces the fastest sensorimotor reflex performed by fish (~5 milliseconds (ms) to initiate, ~10 ms to complete) (*Eaton et al., 1977*; *Fetcho, 1991*; *Burgess and Granato, 2007*; *Satou et al., 2009*). The Mauthner circuit that produces the escape is a combination of electrical, excitatory chemical, and inhibitory chemical synapses contained within a relatively simple set of neurons that must be coordinated to ensure fast and robust escapes (*Fetcho, 1991*). The circuit is wired to ensure that it makes unidirectional turns away from the stimulus. To do so, vibrational stimuli are transmitted to the Mauthner neurons via the VIIIth cranial nerve auditory afferents, which make prominent and stereotyped mixed electrical and glutamatergic chemical synapses at so-called Club Ending synapses, hereafter called the Aud/M synapse (*Pereda et al., 2004*; *Yao et al., 2014*). Mauthner in turn sends a contralateral projection down the length of the spinal cord where it activates primary motorneurons (MNs) in each hemisegment and drives the stereotypical C-bend-shaped escape away from stimulus (*Fetcho, 1991*). The projecting Mauthner axon concurrently activates CoLo interneurons in each segment via excitatory electrical synapses; CoLo in turn re-crosses the spinal cord and inhibits MNs on the stimulus side, thereby ensuring coordinated turns away from the stimulus (*Satou et al., 2009*). Throughout the paper we use the terms pre- and postsynaptic for electrical synapses based on three main criteria: (1) information flow, as described above, from auditory afferents (pre) to Mauthner dendrite (post), then from Mauthner axon (pre) to

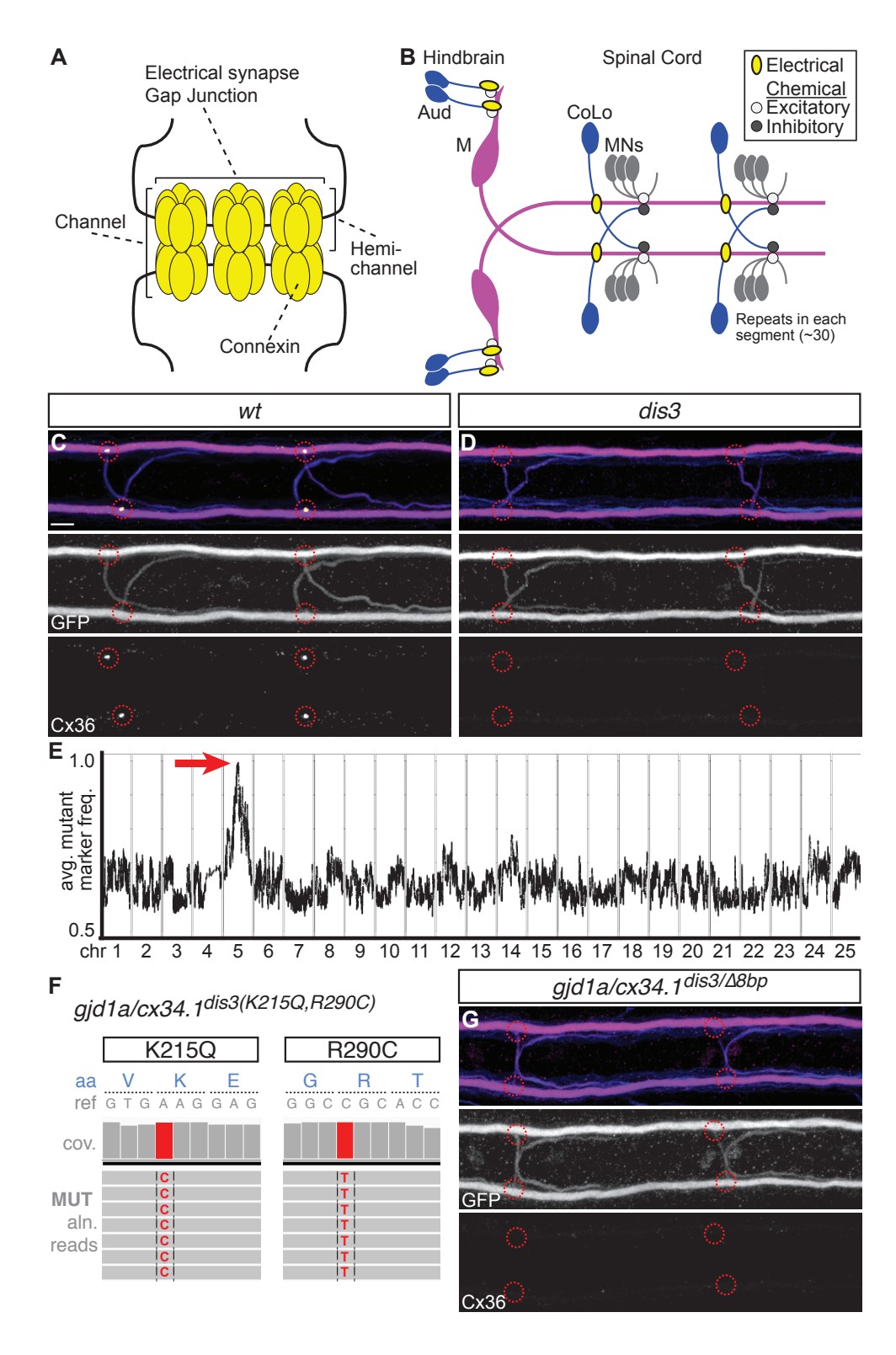

**Figure 1.** *gjd1a/cx34.1* is required for electrical synapse formation. (A) Schematic of an electrical synapse, a specialized neuronal gap junction. (B) Simplified schematic of the zebrafish Mauthner (M) circuit in dorsal view with anterior to the left. Neurons and synapses of the hindbrain and two, of 30, spinal segments are shown. In the hindbrain mixed electrical/chemical synapses are made between auditory (Aud) afferent neurons and the M cell lateral dendrite (Aud/M electrical synapses). In the spinal cord the M axon makes electrical synapses with commissural local (CoLo) interneurons found

*Figure 1 continued on next page*

*Figure 1 continued*

in each segment (M/CoLo electrical synapses). (**C,D**) Two representative dorsal views of spinal cord segments from *M/CoLo:GFP* larvae at 5 days post fertilization. Images are maximum intensity projections of ~10 uM with anterior to the left. Scale bar = 10 uM. Larvae are stained for anti-GFP (magenta), anti-human-Connexin36 (Cx36, yellow), and for neurofilaments (RMO44, blue). Individual GFP and Cx36 channels are shown in neighboring panels. Associated experimental statistics are found in the figure-related table. The Cx36 staining found at wildtype M/CoLo synapses (**C**), red circles) is lost in *dis3* mutants (**D**), red circles). (**E**) Genome wide RNA-seq-based mapping data. The average frequency of mutant markers (black marks) is plotted against genomic position. A single region on chromosome 5 (chr5) emerges with an allele frequency near 1.0 indicating linkage to the *dis3* mutation (red arrow). Each chromosome is separated by vertical lines and labeled at the bottom. (**F**) Mutant reads from the RNA-seq mapping data at two separate positions within the *gjd1a/cx34.1* gene are shown aligned to the reference genome identifying two missense changes within *dis3* mutant animals. Wildtype reference (ref) genome nucleotides and encoded amino acids (aa) are noted. Aligned mutant (MUT) reads are shown as grey boxes; colored letters highlights differences from reference. (**G**) Electrical synapses are lost in trans-heterozygous animals carrying a *dis3* and an 8 bp frameshift allele (*dis3/8 bp)* in *gjd1a/cx34.1*. Images of the spinal cord as in (**C,D**). Associated quantitation of Cx36 at wildtype or mutant synapses can be found in source data for *Figure 1*.

The following source data is available for figure 1:

**Source data 1.** *gjd1a/cx34.1* is required for electrical synapse formation.

CoLo (post) (*Fetcho, 1991*; *Satou et al., 2009*); (2) Mauthner itself is a bipolar neuron with unique axonal (pre) and dendritic (post) compartments (*Kimmel et al., 1981*); (3) both the Aud/M and M/CoLo electrical synapses have closely associated chemical synapse components with the neurotransmitter receptors localizing to postsynaptic compartments of the neurons (Mauthner dendrite in the hindbrain [*Yao et al., 2014*], CoLo proximal portion of the neurite on the ipsilateral side of the spinal cord [*Miller et al., 2015*]). The Mauthner and CoLo neurons can be visualized using the transgenic line *zf206Et(Tol-056)*, hereafter called *M/CoLo:GFP*, which expresses GFP in both neuron types (*Satou et al., 2009*). The electrical synapses of the Mauthner circuit can be visualized by immunostaining using a polyclonal antibody against the human Cx36 protein (*Figure 1C*)(*Miller et al., 2015*).

To identify genes required for electrical synapse formation we mutagenized animals with N-ethyl-N-nitrosourea (ENU) to generate random genomic mutations and created gynogenetic-diploid mutant animals (*Walker et al., 2009*) to screen for disruptions to the Cx36 staining found at M/CoLo synapses at 3 days post fertilization (dpf)(*Miller et al., 2015*). We identified a mutation we called *disconnect3* (*dis3*) that caused a complete loss of Cx36 staining at the M/CoLo synapses (*Figure 1D*). Cx36 staining was disrupted at Mauthner electrical synapses and other electrical synapses in *dis3* mutants across all timepoints examined (2 to 14 dpf). The stereotyped number of Mauthner and CoLo neurons was not disrupted in *dis3* mutants: Mauthner –wildtype, heterozygous, and mutant animals had 2 ± 0 Mauthners / animal; CoLo – wildtype, heterozygous, and mutant avg. 1.975 ± 0.06, 2 ± 0.06, and 2 ± 0 CoLos/8 segments / animal, respectively (segments 5–12 were counted, mean ± standard deviation, n = 5, 11, 5 for wildtype, heterozygous, and mutant, respectively). Moreover, in *dis3* mutants the M and CoLo neurites contact one another in the spinal cord similar to wildtype (*Figure 1C,D*). These results suggest that the gene disrupted by the *dis3* mutation may have a specific role in electrical synapse formation.

To identify the causative gene we used an RNA sequencing (RNA-seq)-based approach that identifies shared regions of genomic homozygosity in a pool of mutant animals (*Miller et al., 2013*). We separately pooled 108 mutant (-/-) and 108 wildtype (+/+ and +/-) siblings and extracted and sequenced mRNA (Illumina Hi-Seq) from each pool. The sequences were aligned to the genome and single nucleotide polymorphisms (SNPs) were identified in the wildtype pool to serve as mapping markers. The SNP frequency was assessed in the mutant pool, identifying an ~1.8 megabase (Mb) region of homozygosity on chromosome 5 in *dis3* mutants (*Figure 1E*). Within this region we used the RNA-seq data to look for potentially deleterious mutations and found that there were two missense changes to a Cx-encoding gene (*Figure 1F*). Cx proteins are labeled for their molecular weight, while the genes encoding the proteins are named for one of five classes in which they reside (*Söhl and Willecke, 2004*). The missense changes identified in *dis3* mutants disrupts the gene *gap junction delta 1a* (*gjd1a*) encoding a Cx with a predicated molecular weight of 34.1 kiloDaltons (Cx34.1). This gene is highly related to the mammalian *gjd2* gene encoding Cx36 (see below). For clarity throughout, when referring to zebrafish Cxs we will use both the official gene name as well as

its molecular weight designation, in this case *gjd1a/cx34.1*. Within the ~1.8 Mb region of mapping there were an additional 10 missense mutations in other genes and one gene with significantly reduced expression compared to wildtype. To determine if the mutations in *gjd1a/cx34.1* were causative for the phenotype we generated an eight base pair (bp) frame-shifting mutation at nucleotide 4 of exon 1 (*gjd1a^fh436^*) using transcription activator-like-effector nucleases (TALENs) (*Sanjana et al., 2012*; *Shah et al., 2015*). We found that larvae homozygous for the frameshift mutation (*gjd1a^fh436/fh436^*) phenocopy the loss of Cx36-staining at M/CoLo synapses observed in the *dis3* mutants; moreover, *gjd1a^fh436^* failed to complement the *dis3* mutation (*Figure 1G*). We renamed the *dis3* mutation *gjd1a^fh360^* and conclude that *gjd1a/cx34.1* is required for M/CoLo electrical synapse formation.

## *gjd1* (Cx34) and *gjd2* (Cx35/Cx36) are part of a family of highly conserved genes/proteins

In mammals there are five unique Connexins that form electrical synapses, with the *gjd2/cx36* gene being the most studied due to its widespread use in the nervous system (*Connors and Long, 2004*; *Söhl et al., 2005*; *Pereda, 2014*). Previous work in zebrafish and goldfish has implicated two Cx36-related proteins, Cx34.7 and Cx35.1, as being present at electrical synapses (*Pereda et al., 2003*; *O'Brien et al., 2004*; *Carlisle and Ribera, 2014*; *Rash et al., 2013*). We wondered how *gjd1a/cx34.1* related to these and other Connexins and so used its sequence to search the zebrafish genome for similar genes. This identified three highly-related loci, two of these being the previously identified *gjd1b/cx34.7* and *gjd2b/cx35.1*, while the third is predicted to encode a Cx with a molecular weight of 35.5 kD, *gjd2a/cx35.5* (*Figure 2*). Within this group of four Cx-encoding genes, all of the predicted proteins share greater than 85% amino acid identity to each other and to mammalian Cx36 proteins (*Figure 2A*). The main variation in protein sequence arises in the intracellular loop, with some modest variation in the C-terminal tail (*Figure 2A,B*). We examined the evolutionary relationship between these proteins and found that they represent two distinct Cx sub-families (*Figure 2C*). In basal vertebrates (lamprey) we found only one gene within the family, this gene was subsequently duplicated creating two sub-families – the Cx35/36 proteins encoded by *gjd2* genes and the Cx34 proteins encoded by genes we have named *gjd1*. Genes within both of these families are found in lineages up to and including birds, but in mammals the *gjd1* family has apparently been lost. In bony fishes, including zebrafish, these gene families underwent another duplication event (*Postlethwait et al., 2004*), resulting in the four extant genes (*Figure 2C*). Previous work has found that the *gjd1b/cx34.7* and *gjd2b/cx35.1* genes are expressed in the nervous system of zebrafish, analogous to *gjd2* (Cx36) expression in mammals (*Li et al., 2009*; *Carlisle and Ribera, 2014*). We examined published RNA-seq and EST datasets (*Ensembl*: WTSI stranded RNA-Seq, WTSI, KIT, Yale [*Yates et al., 2016*] and the UCSC Genome Browser [*Kent et al., 2002*]) and found that all four of the *gjd1* and *gjd2* genes in zebrafish are enriched for nervous system expression while being reduced or absent from other tissues. Together it appears that all of the *gjd1/2* genes have the potential to play broad roles in neural circuit formation and function.

## *gjd1a/cx34.1* and *gjd2a/cx35.5* are required for electrical synapse formation and function

While we identified *gjd1a/cx34.1* as being required for M/CoLo synapse formation (*Figure 1*) we wondered if any of the other closely related genes were required for Mauthner circuit synaptogenesis. To examine these genes we generated mutations in each of them (*Figure 2B*) and analyzed the effect of their loss on electrical synapse formation (*Shah et al., 2015*). We examined the electrical synapses made onto the M cell body and dendrite (*Figure 3A*) and particularly focused on the stereotyped Aud/M synapses (*Figure 3A* arrows, *Figure 3B*) and the M/CoLo electrical synapses of the spinal cord (*Figure 3C*). The anti-human-Cx36 antibody used for staining recognizes all four of the proteins generated by the zebrafish *gjd1/2* genes when each is expressed in HeLa cells (see Materials and methods). We found that *gjd1a/cx34.1* mutants affected not only the M/CoLo synapses in the spinal cord, but also the Aud/M synapses, as well as other electrical synapses made onto the Mauthner cell body (*Figure 3D–F*). In analyzing mutants for the other three genes, we found that *gjd2a/cx35.5* mutants also lost Mauthner electrical synapses in the hindbrain and spinal cord (*Figure 3G–I*) while *gjd1b/cx34.7* and *gjd2b/cx35.1* had no effect on Cx36 staining (*Figure 3J,K*). In quantifying the amount of Cx36-staining at Mauthner synapses we found that in *gjd2a/cx35.5*

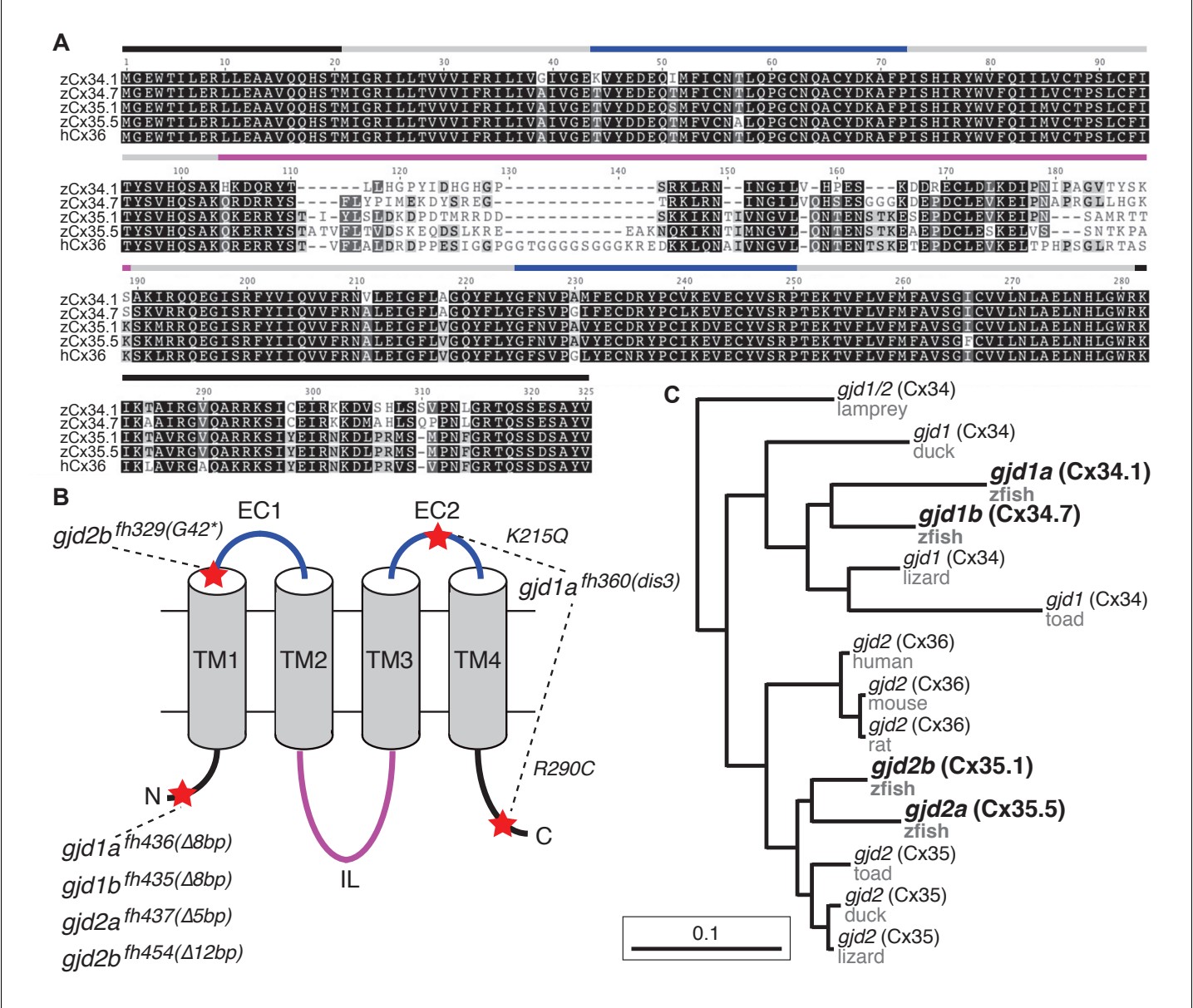

**Figure 2.** The zebrafish genome contains four Connexin-encoding genes related to mammalian *gjd2/cx36*. (**A**) Amino acid alignment of four zebrafish Cx36-like proteins with human Cx36. Colored lines indicate approximate location of the domains shown in the schematic in B. (**B**) Schematic of Cx36-like proteins. Blue: extracellular (EC) loops; grey cylinders: transmembrane (TM) domains; black: intracellular N- and C-terminal tails; magenta: intracellular (IL) loop. Red stars indicate the positions of mutations in ENU-induced (*fh329*, *fh360*) and engineered (*fh435*, *fh436*, *fh437*, *fh454*) mutations in the four zebrafish genes encoding Cx36-like proteins: *gjd1a/cx34.1*, *1b/cx34.7*, *2a/cx35.5*, and *2b/cx35.1*. Details of mutations can be found in the figure-related table. (**C**) Phylogeny of vertebrate Cx36 proteins. *gjd2a* and *2b* are duplicates of the single mammalian gene *gjd2*, which encodes Cx36. *Gjd1a* and *1b* are duplicates of a gene that was lost in the tetrapod lineage. Lamprey = *Petromyzon marinus*, duck = *Anas platyrhynchos*, lizard = *Anolis carolinensis*, toad = *Xenopus tropicalis*, mouse = *Mus musculus*, rat = *Rattus norvegicus*, human = *Homo sapiens*, zfish = *Danio rerio*. Sequence information of mutants can be found in source data for *Figure 2*.

The following source data is available for figure 2:

**Source data 1.** The zebrafish genome encodes four Connexin-encoding genes related to mammalian *gjd2/cx36*.

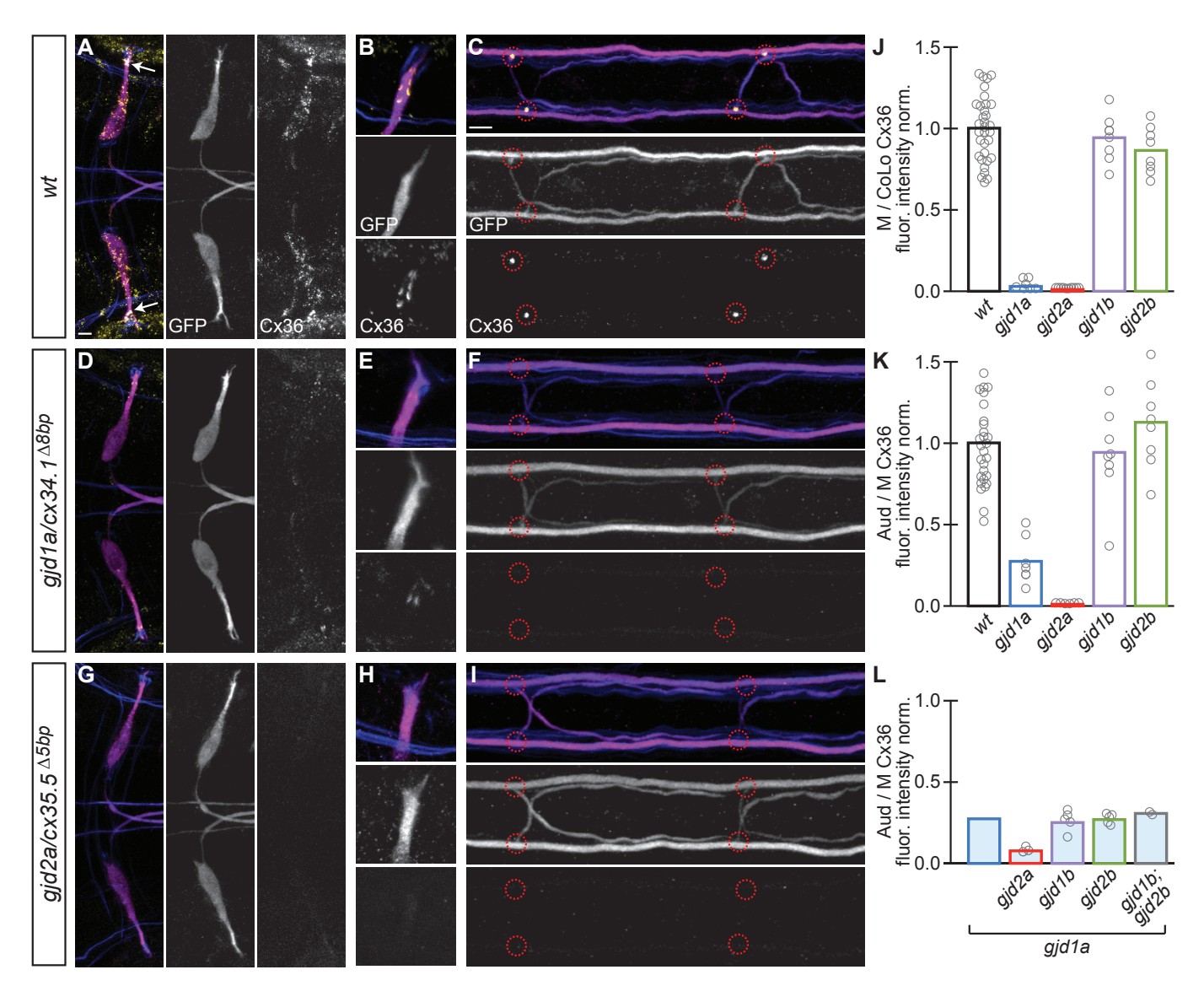

**Figure 3.** Electrical synapses of the Mauthner circuit require both *gjd1a/cx34.1* and *gjd2a/cx35.5*. (A–I). In this and all subsequent figures, unless otherwise specified, images are dorsal views of hindbrain, Mauthner lateral dendrite, and two spinal cord segments from *M/CoLo:GFP* larvae at 5 days post fertilization. Hindbrain, lateral dendrite, and spinal cord images are maximum intensity projections of ~30,~5, and ~10 uM, respectively. Anterior is to the left. Scale bar = 10 uM. Larvae are stained for anti-GFP (magenta), anti-human-Connexin36 (Cx36, yellow), and for neurofilaments (RMO44, blue). Individual GFP and Cx36 channels are shown in neighboring panels. Electrical synapses are found on the Mauthner cell body (A) with prominent and stereotyped Aud/M synapses found on Mauthner's lateral dendrite (arrows in A), (B) and at M/CoLo synapses in the spinal cord (C), red circles). (D–I) In *gjd1a/cx34.1* and *gjd2a/cx35.5* mutants Mauthner electrical synapses are lost or, in the specific case of Aud/M synapses in *gjd1a/cx34.1* mutants, diminished. (J–L) Bar graphs represent the mean of the indicated value quantified at synapses with each circle representing the average of 12–16 M/CoLo or 8–12 Aud/M synapses within an animal. (J,K) M/CoLo electrical synapses are absent in *gjd1a/cx34.1* and *gjd2a/cx35.5* mutants, while Aud/M club endings are strongly reduced in *gjd1a* mutants and absent in *gjd2a* mutants. *gjd1b* and *gjd2b* have no effect on Mauthner electrical synapses. N = L. The remaining Cx36 staining observed at Aud/M synapses in *gjd1a* mutants is not due to *gjd1b* or *gjd2b*. For reference, the first bar is a duplication of *gjd1a* mutant data from K. In *gjd1a;gjd2a* double mutants the remaining Cx36 staining is lost as expected given that *gjd2a* is required for Aud/M synapses (K). In double and triple mutants combinations between *gjd1a*, *gjd1b*, and *gjd2b* there is no effect on the remaining Cx36 staining at Aud/M synapses. Associated quantitation of Cx36 at wildtype or mutant synapses can be found in *Figure 3—source data 1* for *Figure 3* and *Figure 3—source data 2* for *Figure 3*.

The following source data is available for figure 3:

**Source data 1.** Electrical synapses of the Mauthner circuit require both *gjd1a/cx34.1* and *gjd2a/cx35.5*.

*Figure 3 continued*

**Source data 2.** Electrical synapses of the Mauthner circuit require both *gjd1a/cx34.1* and *gjd2a/cx35.5*.

mutants there was a complete absence of detectable Cx protein at synapses throughout the Mauthner circuit. In *gjd1a/cx34.1* mutants the Cx36 staining at M/CoLo synapses was lost, but we note that ~ 30% of staining remained (an ~4 fold reduction) at the Aud/M synapses (*Figure 3E,J,K*). This residual staining was not eliminated in double or triple mutants between *gjd1a/cx34.1* and *gjd1b/cx34.7 or gjd2b/cx35.1* (*Figure 3L*). On the whole we conclude that the gap junctions that form the Mauthner circuit electrical synapses require both *gjd1a/cx34.1* and *gjd2a/cx35.5* for their formation.

To investigate whether Cx34.1 and Cx35.5 were required for electrical synapse function we examined whether the passage of the gap junction permeable dye neurobiotin (Nb) was impaired in mutants. We retrogradely labeled Mauthner axons with Nb from caudal spinal cord transections (*Figure 4A*) and then detected Nb within the CoLo cell bodies in the *M/CoLo:GFP* line (*Figure 4B*) (*Miller et al., 2015*). In both *gjd1a/cx34.1* and *gjd2a/cx35.5* mutants we found that no Nb was transferred from Mauthner to CoLo (*Figure 4C–G*). By contrast, we found that in *gjd1b/cx34.7* and *gjd1b/cx35.1* mutants Nb transfers from Mauthner to CoLo similar to wildtype (*Figure 4G*). That Nb passage is completely blocked in both *gjd1a/cx34.1* and *gjd2a/cx35.5* mutants supports the idea that Mauthner circuit electrical synapses are dependent on both Cx34.1 and Cx35.5 proteins for their function. Taken together these data suggest that there is an intimate interaction between these two Cxs required for the establishment of M electrical synapses.

## *gjd1a* (Cx34.1) and *gjd2a* (Cx35.5) are localized at and required for the majority of electrical synapses in the hindbrain and spinal cord

To examine whether Cx34.1 and Cx35.5 are both present at electrical synapses we generated antibodies that specifically recognized either protein (see Materials and methods for details). Both antibodies generated staining patterns highlighting the same structures in the hindbrain and spinal cord as staining with the antibody against human Cx36 (*Figure 5A–C*). Cx34.1 and Cx35.5 staining overlapped nearly completely throughout the hindbrain and spinal cord, but there were rare examples where staining was apparent for only one of the two Cx proteins (*Figure 5A*). At Mauthner circuit electrical synapses Cx34.1 and Cx35.5 staining was completely overlapping at the individually identifiable Aud/M and M/CoLo synapses in the hindbrain and the spinal cord (*Figure 5A–C*). In *gjd1a/cx34.1* mutants we found that Cx34.1 staining was lost while in the *gjd2a/cx35.5* mutants we found that Cx35.5 staining was lost supporting both the specificity of the antibodies as well as the deleterious nature of the mutations (*Figure 5D–I*). In *gjd2a/cx35.5* mutants we found that Cx34.1 staining was lost from Aud/M and M/CoLo synapses suggesting Cx34.1 requires Cx35.5 for its localization at Mauthner synapses (*Figure 5G–I*). In *gjd1a/cx34.1* mutants we found that Cx35.5 was completely lost from M/CoLo synapses (*Figure 5F*) and residual Cx35.5 staining was found at Aud/M synapses similar to what was observed with the human Cx36 antibody (*Figure 5D,E*; *Figure 3D,E,J,K*). Additionally, in both *gjd2a/cx35.5* and *gjd1a/cx34.1* mutants we detected dim Cx34.1 and Cx35.5-positive puncta, respectively, that were not associated with the Mauthner circuit (*Figure 5D–I*). Such staining may represent synapses at which Cx34.1 and Cx35.5 do not require one another for their localization. Altogether our results suggest that both Cx34.1 and Cx35.5 are broadly expressed and used throughout the hindbrain and spinal cord of zebrafish. Moreover, while each Cx can apparently be used independently, where Cx34.1 and Cx35.5 colocalize their mutual presence supports the robust localization of the other at electrical synapses.

## Mauthner circuit electrical synapses require *gjd1a/cx34.1* in postsynaptic neurons and *gjd2a/cx35.5* in presynaptic neurons

Since we found that (*gjd1a*) Cx34.1 and (*gjd2a*) Cx35.5 are colocalized and support one another's localization at Mauthner synapses we wondered where the two Cxs were required for electrical synapse assembly. Given that electrical synapses are composed from Cx hemichannels contributed by apposing neurons (*Figure 1A*), both Cx34.1 and Cx35.5 could be found on both sides of the synapse

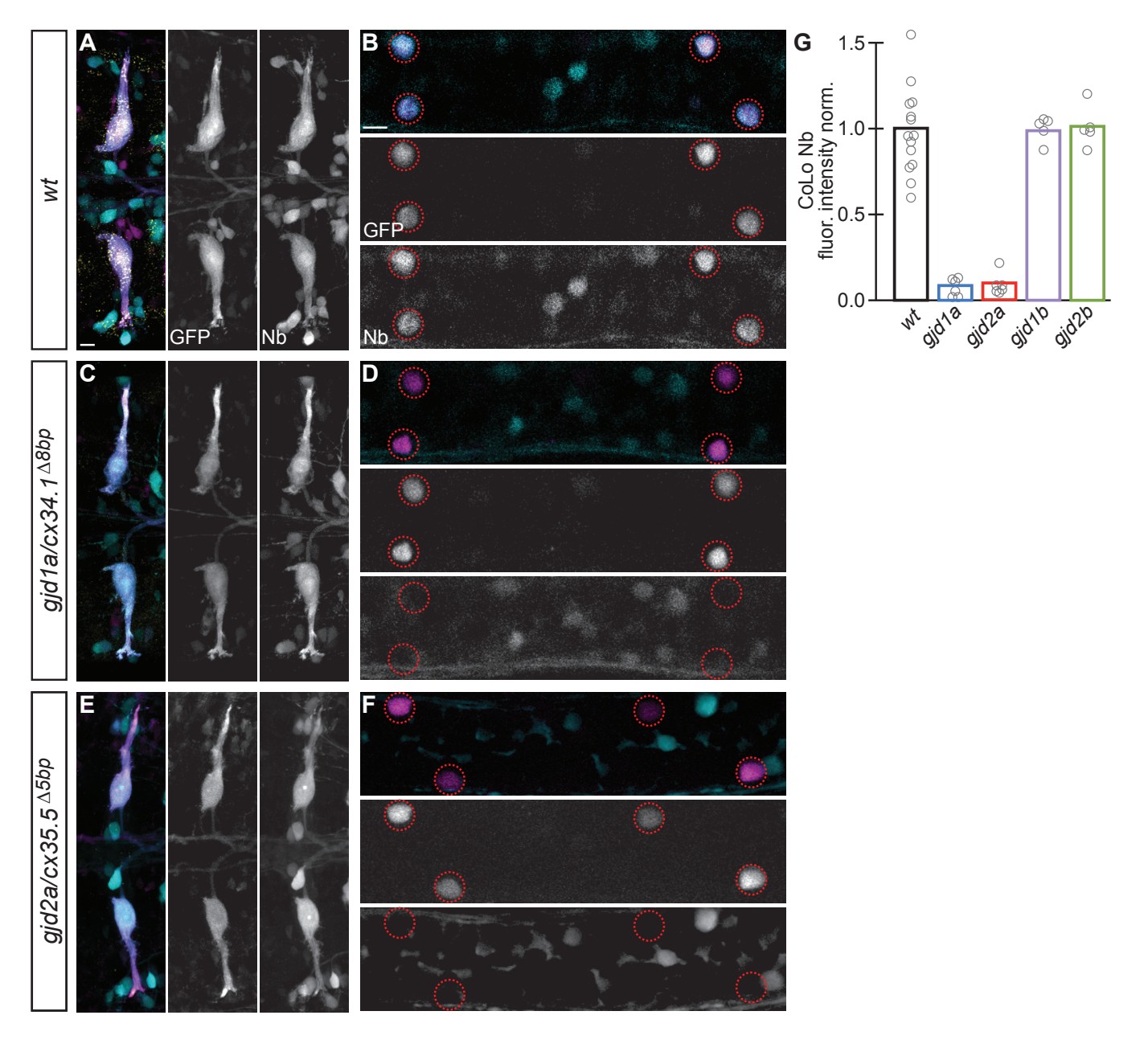

**Figure 4.** Electrical synapses are functionally defective in *gjd1a/cx34.1* and *gjd2a/cx35.5* mutants. Retrograde labeling of Mauthner axons with the gap junction permeable dye Neurobiotin (Nb) from a caudal transection. Hindbrain and spinal cord images are maximum intensity projections of ~30 and~10 uM, respectively. Anterior is to the left. Scale bar = 10 uM. Spinal cord images are at the level of the CoLo cell bodies (circles), which is dorsal to the synapses. (A–F) Larvae are stained for anti-GFP (magenta), biotin (Nb, cyan), and anti-human-Connexin36 (Cx36, yellow). Nb labels the Mauthner cell bodies and other caudally projecting neurons in the hindbrain (A) and passes from the Mauthner axon through the electrical synapses to fill the CoLo cell bodies (B), circles. Other neurons are also labeled due to projections caudally into the lesion site (A, non-Mauthner neurons, B, non-circled cell bodies). (C–F) In *gjd1a/cx34.1* and *gjd2a/cx35.5* mutants Nb labels M normally (C,E) however none passes into CoLos (D,F). (G) Quantitation of the ratio of Nb in CoLo to M cell bodies in wildtype and mutants. Each circle represents the average Nb fluorescence within 8–12 CoLo cell bodies compared to the 2 Mauthner cell bodies in an animal. Associated experimental statistics are found in the figure-related table. Associated quantitation of Nb transfer in wildtype or mutant can be found in source data for *Figure 4*.

The following source data is available for figure 4:

**Source data 1.** Electrical synapses are functionally defective in *gjd1a/cx34.1* and *gjd2a/cx35.5* mutants.

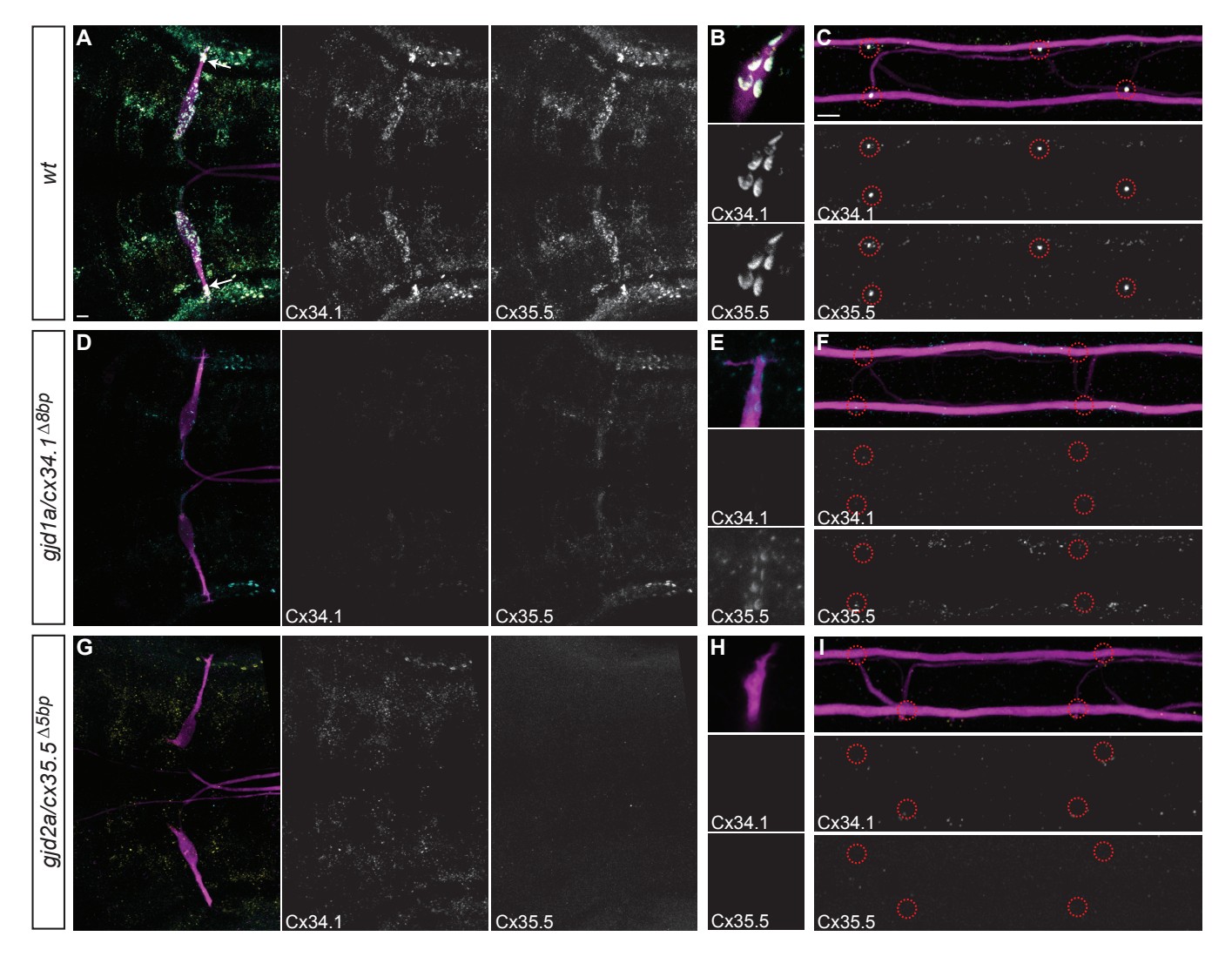

**Figure 5.** Recruitment of Gjd1a/Cx34.1 and Gjd2a/Cx35.5 to electrical synapses is co-dependent. Larvae are stained for anti-GFP (magenta), anti-Connexin34.1 (Cx34.1, yellow), and for anti-Connexin35.5 (Cx35.5, cyan). Individual Cx34.1 and Cx35.5 channels are shown in neighboring panels. Hindbrain, lateral dendrite, and spinal cord images are maximum intensity projections of ~30,~5, and ~10 uM, respectively. Anterior is to the left. Scale bar = 10 uM. (A–C) Cx34.1 and Cx35.5 are found colocalized at electrical synapses throughout the hindbrain (A) including at Aud/M (arrows in A, (B), and M/CoLo synapses (C). In *gjd1a/cx34.1* mutants the localization of Cx35.5 to electrical synapses is lost (D–F) or diminished at Aud/M synapses (E); in *gjd2a/cx35.5* mutants the localization of Cx34.1 to electrical synapses is lost (G–I). Associated antibody information can be found in source data for *Figure 5*.

The following source data is available for figure 5:

**Source data 1.** Recruitment of Gjd1a/Cx34.1 and Gjd2a/Cx35.5 to electrical synapses is co-dependent.

(heteromeric hemichannels), or instead each could be found exclusively on one side or the other (heterotypic channels). To test where each gene was required for electrical synapse formation we performed cell transplants at the blastula stage (*Kemp et al., 2009*) to create chimaeric animals containing wildtype and mutant cells and examined in which neuron of the circuit (auditory, Mauthner, or CoLo) the genes were required. We focused on the Mauthner circuit because we are able to associate the stereotyped synapses of the circuit with the uniquely identifiable pre- and postsynaptic neurons (see *Figure 1B* and associated discussion). To create chimaeric animals we transplanted

cells at the blastula stage from donor embryos marked with either the *M/CoLo:GFP* transgene or Biotin Dextran (BD) into host embryos lacking the same marker and examined the resulting embryos at 5 dpf (*Figure 6*). To mark the presynaptic auditory afferent neurons and examine the hindbrain Aud/M synapses we transplanted cells from non-transgenic, BD-injected embryos into an *M/CoLo: GFP* host. To mark the postsynaptic Mauthner neurons in experiments examining the Aud/M synapse, and in all experiments in the spinal cord examining M/CoLo synapses, we transplanted cells from *M/CoLo:GFP* transgenic donors into a non-transgenic host. Such chimaeric animals allowed us to unambiguously identify from which embryo the neurons of the Mauthner circuit were derived based on the presence or absence of the markers used in each experiment.

In control chimeras with donor-derived presynaptic or postsynaptic neurons we saw no effect on Cx36 staining at Aud/M or M/CoLo synapses (*Figure 6A–E*). Similarly, when presynaptic neurons (auditory afferent neurons in the hindbrain [*Figure 6F*] or Mauthner in the spinal cord [*Figure 6I*]) lacked *gjd1a/cx34.1* the electrical synapses were normal. However, when the postsynaptic neuron (Mauthner in the hindbrain [*Figure 6G,H*] or CoLo in the spinal cord [*Figure 6J*]) lacked *gjd1a/cx34.1* the electrical synapses were affected. Conversely, we found exactly the opposite requirement for *gjd2a/cx35.5*: removing its function presynaptically, but not postsynaptically, resulted in the loss of associated electrical synapses (*Figure 6K–O*). Moreover, presynaptic removal of *gjd2a/cx35.5* resulted in a complete loss of Cx36 staining at any associated synapse, while postsynaptic loss of *gjd1a/cx34.1* resulted in no staining at M/CoLo synapses and an ~4 fold reduction of Cx36 staining at Aud/M synapses (*Figure 6P,Q*), consistent with the decrease observed in whole mutant animals (*Figure 3J,K*). This suggests that at M/CoLo synapses Cx35.5 and Cx34.1 are the exclusive pre- and postsynaptic Cxs, respectively, while at the Aud/M synapses Cx35.5 is the sole presynaptic Cx while Cx34.1 is required postsynaptically for the majority of GJ channels (*Figure 6P,Q*). We note that even within the single Mauthner neuron both Cx-encoding genes are required for electrical synapse formation, but their functions are spatially restricted to the dendrite (*gjd1a/cx34.1*, *Figure 6G,H*) versus the axon (*gjd2a/cx35.5*, *Figure 6N*). We conclude that *gjd2a/cx35.5* and *gjd1a/cx34.1* are required pre- and postsynaptically, respectively, at Mauthner circuit electrical synapses.

The above transplant experiments suggest that some GJ channels at Mauthner electrical synapses are heterotypic, with unique Cxs used on each side of the synapse. Moreover, our results suggest that a trans-synaptic interaction between the Cxs is required for stabilization, and thereby detection via antibody staining, at the Mauthner synapses. This would help explain why when we remove either *gjd2a/cx35.5* or *gjd1a/cx34.1* we observe reduced staining of the other Cx at the synapse. This model predicts that in a *gjd2a/Cx35.5* mutant the postsynaptic Cx34.1 should still be available to make electrical synapses if provided with an appropriate presynaptic Cx (and vice versa). To test this notion and ask if Cx35.5 or Cx34.1 are sufficient pre- and postsynaptically, respectively, we created chimeric animals as above except we transplanted from wildtype *M/CoLo:GFP* embryos into a non-transgenic mutant host (*Figure 7*). We found that when the postsynaptic neuron was wildtype in a *gjd1a/cx34.1* mutant host the electrical synapses were rescued (*Figure 7A,B,D*); however, when the presynaptic neuron was wildtype there was no rescue (*Figure 7C*). Conversely, a wildtype presynaptic neuron was sufficient to rescue the synapse in a *gjd2a/cx35.5* mutant (*Figure 7G*), but had no effect when the wildtype neuron was postsynaptic (*Figure 7E,F,H*). In other words, Cx35.5 is sufficient to rescue the synapse if it is present in the presynaptic neurons while Cx34.1 is sufficient to rescue the synapse if it is present in the postsynaptic neuron. We note that the fold-rescue of Cx36 staining by a wildtype postsynaptic neuron was greater at M/CoLo as compared to Aud/M synapse (*Figure 7I,J*); this ~4 fold rescue is consistent with the fold reduction seen in *gjd1a* mutants and postsynaptic loss of *gjd1a* at these same synapses (see *Figure 3J,K* and *Figure 6P,Q*). Altogether we conclude that electrical synapse formation in the Mauthner circuit requires presynaptic Cx35.5 and postsynaptic Cx34.1 and that robust Cx localization at the synapse requires a trans-synaptic interaction with an appropriate Cx in an adjacent cell.

## Electrical synapses are required for speed and coordination of the Mauthner-induced escape response

To test how electrical synapses contributed to the escape response we examined the behavior of mutant and wildtype siblings using high-throughput behavioral analysis in free-swimming 6 dpf larvae (*Wolman et al., 2015*). The Mauthner circuit generates a fast escape response away from threatening stimuli using a combination of both electrical and chemical synapses (*Fetcho, 1991*); for an

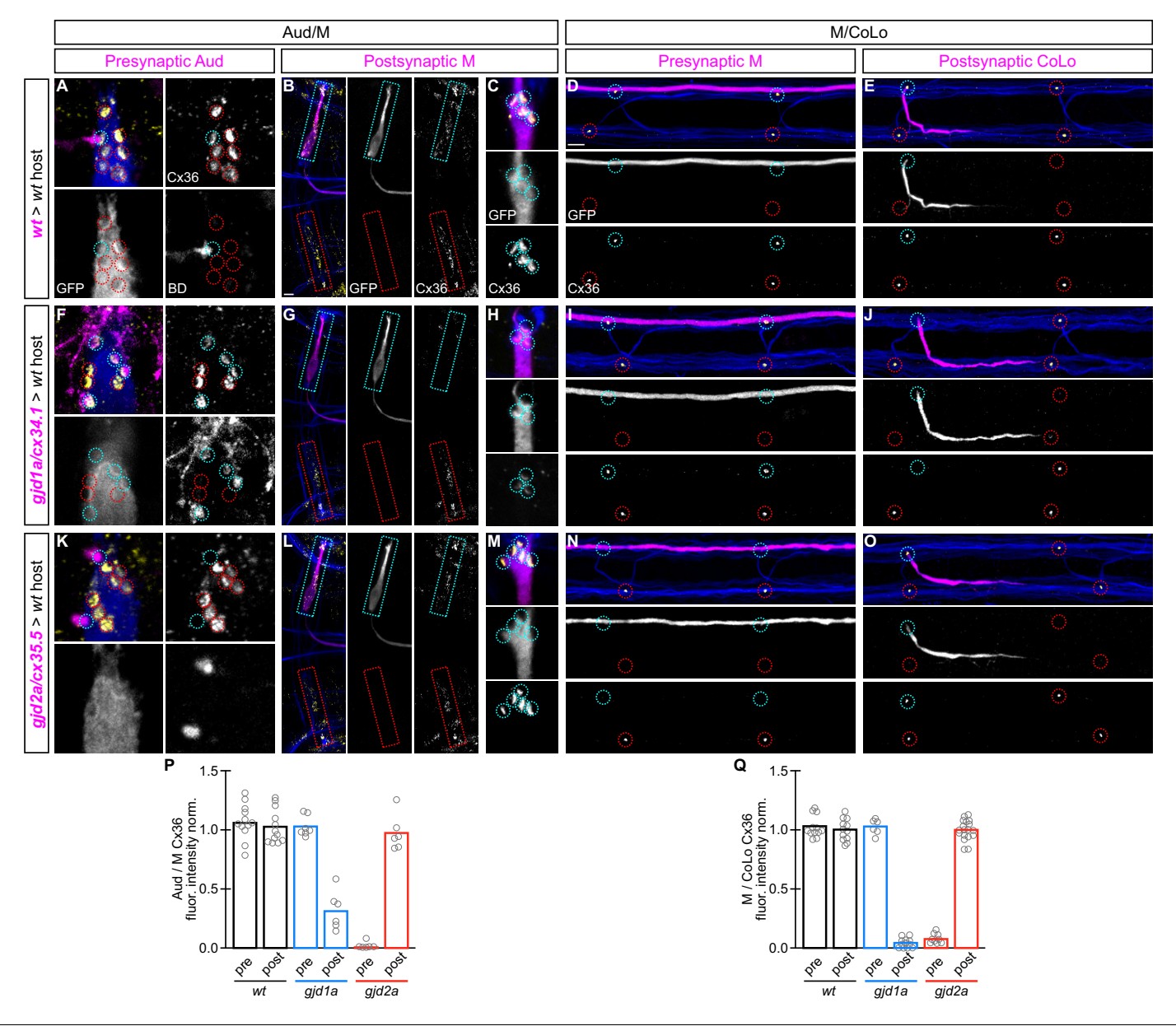

**Figure 6.** *gjd1a/cx34.1* and *gjd2a/cx35.5* are required asymmetrically at Mauthner electrical synapses. Dorsal views of chimeric larvae containing Biotin-Dextran- (BD) or GFP-marked cells transplanted from a donor embryo of noted genotype into a wildtype (wt) host; throughout the figure the neurons derived from the donor embryo are displayed in magenta, while those from the host are in blue. Synapses (stained with anti-human-Cx36, yellow) associated with a transplanted neuron (cyan circles and boxes) can be directly compared to wildtype host synapses (red circles and boxes). Hindbrain, lateral dendrite, and spinal cord images are maximum intensity projections of ~30,~5, and ~10 uM, respectively. Anterior is to the left. Scale bar = 10 uM. (A–C) At the Aud/M synapses presynaptic auditory afferent neurons (in A, stained with BD, magenta) synapse onto the postsynaptic Mauthner lateral dendrite (stained with anti-GFP, blue in A, magenta in B,C). (D,E) At the M/CoLo synapses the presynaptic Mauthner axon (stained with anti-GFP, magenta in D) synapses with the postsynaptic CoLo (stained with anti-GFP, magenta in E). (F–J) Presynaptic removal of *gjd1a/cx34.1* function has no effect on Aud/M and M/CoLo synapses (cyan circles in F,I). By contrast, removing *gjd1a/cx34.1* postsynaptically causes a loss of electrical synapse staining (cyan boxes and circles in G,H,J; note residual Cx36 staining at Aud/M synapses when *gjd1a/cx34.1* is removed from only the postsynaptic neuron, (H). (K–O) Conversely, *gjd2a/cx35.5* function is required exclusively presynaptically (K,N) and is dispensable postsynaptically (L,M,O). (P,Q) Quantitation of the ratio of Cx36 at donor-associated synapses to wildtype host synapses in chimaeric embryos. Each circle represents the average ratio of 1–8 donor-associated to 8–12 host-associated synapses within an animal, varying depending on the synapse and chimaera. Associated quantitation of Cx36 at chimeric synapses can be found in *Figure 6—source data 1* for *Figure 6* and *Figure 6—source data 2* for *Figure 6*.

The following source data is available for figure 6:

*Figure 6 continued on next page*

*Figure 6 continued*

**Source data 1.** *gjd1a/cx34.1* and *gjd2a/cx35.5* are required asymmetrically at Mauthner electrical synapses.

**Source data 2.** *gjd1a/cx34.1* and *gjd2a/cx35.5* are required asymmetrically at Mauthner electrical synapses.

explanation of Mauthner circuit function see *Figure 1B* and associated discussion above. At 6 dpf there are approximately six auditory afferent neurons making mixed electrical/glutamatergic chemical synapses onto Mauthner (Aud/M); the electrical component of the postsynaptic response occurs 1–3 ms after afferent stimulation and happens just prior to that of its glutamatergic counterpart (*Yao et al., 2014*). By contrast, electrophysiology at the M/CoLo synapse suggests a single, excitatory electrical coupling allowing Mauthner to activate CoLo during an escape (*Satou et al., 2009*). Thus, electrical synapses are suggested to have unique effects on the initiation (Aud/M) and coordination (M/CoLo) of the Mauthner-induced escape response.

To test mutant behavior animals were placed into individual chambers in a multi-well testing stage illuminated by infrared light and were subjected to startling vibrational (acoustic) stimuli. Behavior was monitored with a high speed camera (1000 frames per second) to capture the kinematics, or body movements, of responses. Each animal and its behavioral response was analyzed using FLOTE software (*Burgess and Granato, 2007*), which automatically tracks movement parameters of the larvae. FLOTE can distinguish between Mauthner-induced short latency C-bend (SLC) escape responses and the longer latency escapes generated by related but distinct circuits (*Burgess and Granato, 2008*). We analyzed the kinematics of the 6 dpf larval escape response from crosses between heterozygous animals (incross of *gjd1a/cx34.1$^{+/-}$* or *gjd2a/cx35.5$^{+/-}$*). Behavioral testing and analysis was performed blind followed by genotyping of each larva and comparisons of responses between wildtype siblings and mutants. We found that mutants produced SLCs at similar frequencies compared to their wildtype siblings (*Figure 8A*, note that we found no difference between wildtype siblings from the *gjd1a/cx34.1$^{+/-}$* or *gjd2a/cx35.5$^{+/-}$* crosses and so have collapsed these two groups into a single wt data point in *Figure 8*; individual statistics can be found in the source data and lead to the same conclusions). However, we found that both *gjd1a/cx34.1* and *gjd2a/cx35.5* mutants initiated escape responses ~2 ms (40%) slower than their wildtype siblings (*Figure 8B*). In these 'slow' SLC responses the kinematics of the escape often had normal turn angles, maximum angular velocity, and other parameters known to be associated with Mauthner induced escapes, albeit happening later than in wildtype (*Figure 8C–H*). However, a subset of mutant responses (~20% in each genotype) occurred with an abnormally shallow maximum angle of the C-bend (*Figure 8C*, arrow) as well as abnormally slow maximum angular velocities (*Figure 8D*, arrow). These abnormal responses suggested a defect in performing the stereotyped C-bend shape of the escape response and so we reanalyzed the video data from these events and found that both *gjd1a/cx34.1* and *gjd2a/cx35.5* mutants produced abnormal postural responses to the auditory startle stimuli (*Figure 8I–L*). We found that as these animals initiated movements they displayed only slight bends to one side, creating 'kinked' or 'S-shaped' postures, and also shortening their body axis. These animals would stay in this state for 4–10 ms then produce a secondary movement in the opposite direction of the initial movement, followed most often by swimming behavior. These secondary movements were likely the normal counter bend and swimming motions that occur after a Mauthner induced escape (*Burgess and Granato, 2008*; *Satou et al., 2009*), but in these cases they occurred after a failed escape. From these data we conclude that electrical synapses contribute to the speed and coordination of the Mauthner induced escape response.

## Discussion

Our genetic analyses in the zebrafish Mauthner circuit support a model where these vertebrate electrical synapses contain molecularly asymmetric gap junctions with unique pre- and postsynaptic Cxs (*Figure 9A*). One intriguing possibility is that differences in the pre- and postsynaptic Cxs would bias the passage of ionic flow through the GJ creating a rectified electrical connection (*Palacios-Prado et al., 2014a*). Indeed such is the case for two Innexins, Shaking-B(Neural +16) and Shaking-B (lethal), that in *Drosophila* form a molecularly asymmetric synapse; when these hemi-channels are

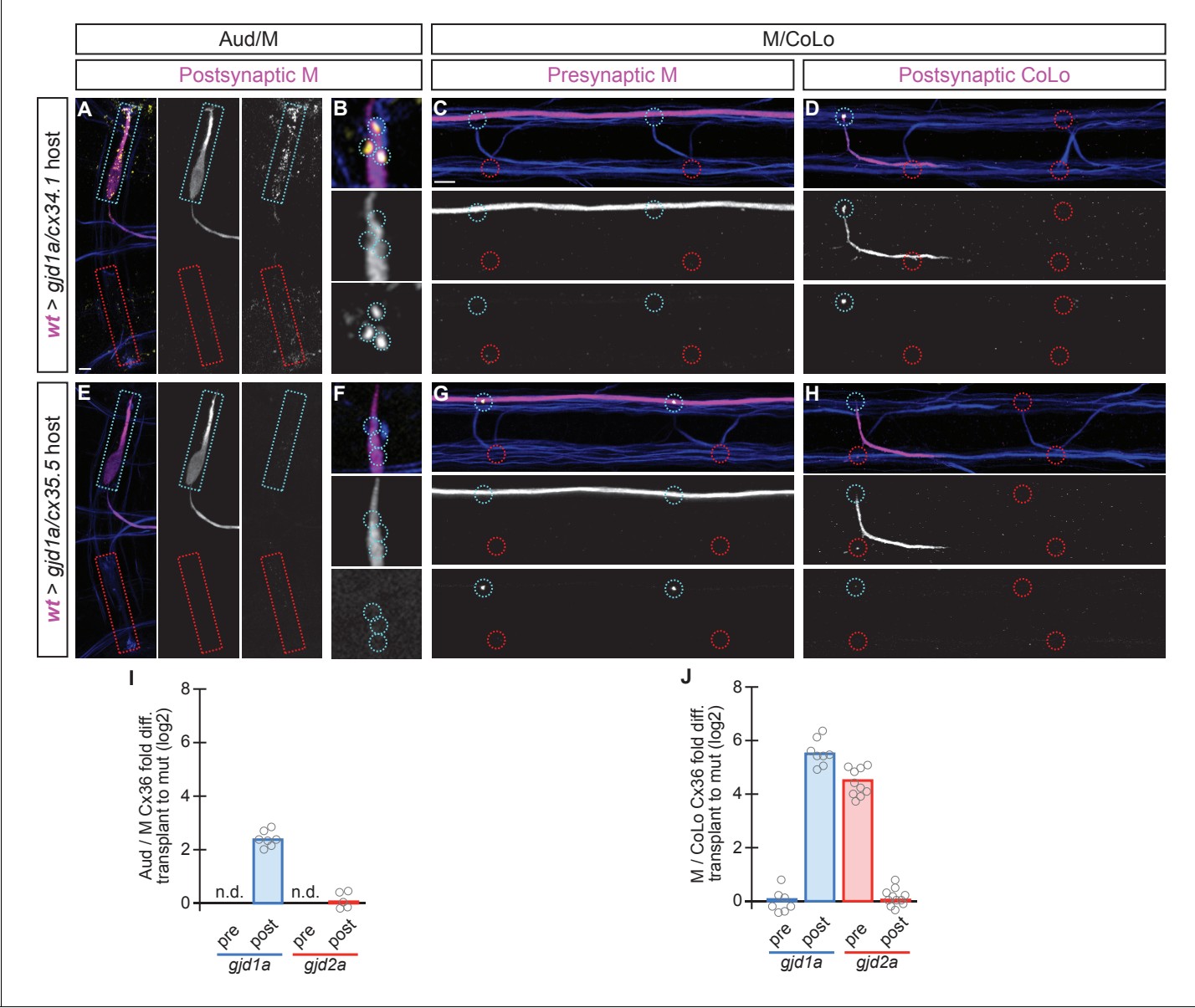

**Figure 7.** The exclusive asymmetric functions of *gjd1a/cx34.1* and *gjd2a/cx35.5* are sufficient for electrical synapse formation. Dorsal views of chimeric larvae containing GFP-marked cells transplanted from a wildtype (wt) donor embryo into a mutant host of noted genotype; throughout the figure the neurons derived from the wt donor embryo are displayed in magenta, while those from the mutant host are in blue. Synapses (stained with anti-human-Cx36, yellow) associated with a transplanted neuron (cyan circles and boxes) can be directly compared to mutant host synapses (red circles and boxes). Hindbrain, lateral dendrite, and spinal cord images are maximum intensity projections of ~30, ~5, and ~10 uM, respectively. Anterior is to the left. Scale bar = 10 uM. (**A–D**) When the postsynaptic neuron of the Aud/M or M/CoLo synapse is wildtype in a *gjd1a/cx34.1* mutant the electrical synapses are rescued (**A,B,D**). By contrast, when the presynaptic neuron is wildtype in a mutant there is no effect on Cx36 staining at the synapse (**C**). (**E–H**) Conversely, electrical synapses are rescued in *gjd2a/cx35.5* mutants only when the presynaptic, and not the postsynaptic, neuron is wildtype. (**I,J**) Quantitation of the fold increase in the ratio of Cx36 at wt-donor-associated synapses to mutant host synapses in chimaeric embryos. Each circle represents the average ratio of 1–8 donor-associated to 8–16 host-associated synapses within an animal. Associated quantitation of Cx36 at chimeric synapses can be found in *Figure 7—source data 1* for *Figure 7* and *Figure 7—source data 2* for *Figure 7*.

The following source data is available for figure 7:

**Source data 1.** The exclusive asymmetric functions of *gjd1a/cx34.1* and *gjd2a/cx35.5* are sufficient for electrical synapse formation.

**Source data 2.** The exclusive asymmetric functions of *gjd1a/cx34.1* and *gjd2a/cx35.5* are sufficient for electrical synapse formation.

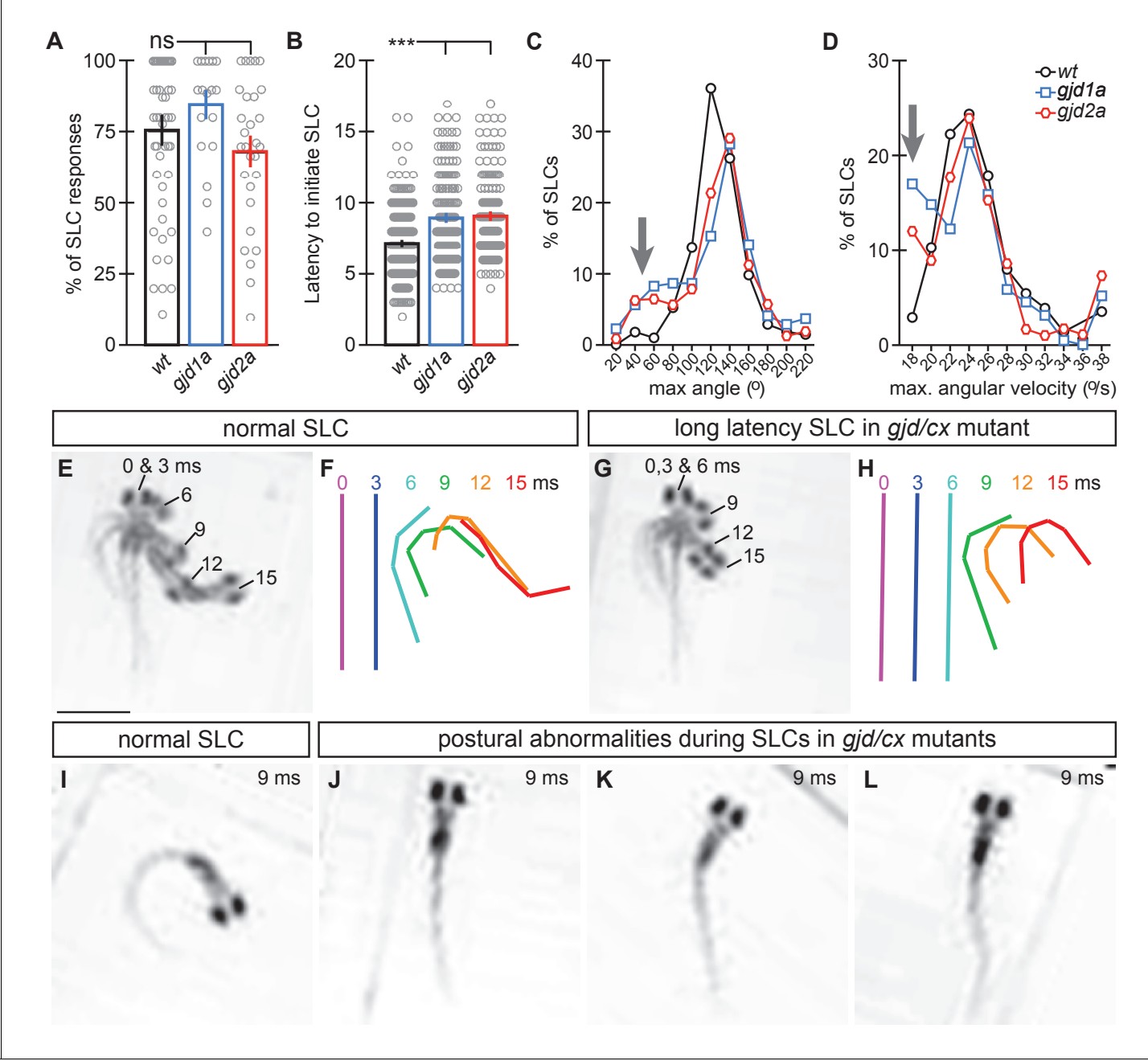

**Figure 8.** *gjd1a/cx34.1* and *gjd2a/cx35.5* mutants have delayed and abnormal escape responses. M-induced escape responses (Short Latency C-bends, SLCs) to a startling vibrational (sound) stimuli executed by 6 day post fertilization larvae were analyzed by high-speed (1000 frame per second) videomicroscopy. Scale bar = 1 mM. (**A**) Frequency of elicited SLCs in wildtype (wt) and indicated mutants in 10 trials per animal. Note that the FLOTE analysis software removes some trials if they cannot be classified. Bar graphs represent data as mean ± SEM with each circle representing an individual animal's average % of response. Mutant larvae execute escapes as frequently as WT (n = 52, 20, and 29 larvae for wt, *gjd1a/cx34.1*, and *gjd2a/cx35.5*, respectively; 1 way ANOVA not significant (ns), Dunn's Multiple Comparison Test: wt to *gjd1a/cx34*.1 and wt to *gjd2a/cx35.5* both ns). (**B**) Latency of elicited SLCs in all individual trials. Bar graphs represent data as mean ± SEM with each circle representing individual SLC latencies. Mutant larvae are significantly delayed in their latency to initiate an M-induced SLC (n = 359, 160, and 180 SLCs from 52, 20, and 29 larvae from wt, *gjd1a/cx34.1*, and *gjd2a/cx35.5*, respectively; 1 way ANOVA p<0.0001, Dunn's Multiple Comparison Test: wt to *gjd1a/cx34*.1 and wt to *gjd2a/cx35.5* both significant at p<0.001). (**C,D**). Kinematic analysis of the maximum SLC turn angle (**C**) and angular velocity (**D**) plotted as the average number of events within an indicated bin. Arrows indicate shallow angle and low velocity turns exhibited by mutants. (**E–H**) Time-lapse analysis of a normal (**E,F**) and delayed (**G,H**) M-induced escape response (SLCs). Individual snapshots taken at the indicated times (ms = milliseconds) are overlaid on an individual image (**E,G**). A line representing the midline body axis at each time was drawn by hand to indicate the movement (**F,H**). (**I–L**) A normal escape bend at its maximum

*Figure 8 continued on next page*

*Figure 8 continued*

angle (I) compared to abnormally shaped escape bends executed by *gjd1a* (J,K) or *gjd2a* (L) mutant larvae. Associated experimental statistics can be found in source data for *Figure 8*.

The following source data is available for figure 8:

**Source data 1.** *gjd1a/cx34.1* and *gjd2a/cx35.5* mutants have delayed and abnormal escape responses.

paired in apposing *Xenopus* oocytes they allow for biased current flow (*Phelan et al., 2008*). In adult goldfish freeze-fracture immunolabeling and electron microscopy (EM) suggest that the Aud/M (club ending) synapses of Mauthner are molecularly asymmetric (see below for discussion of Cx usage) and electrophysiology found that ionic flow was preferentially directed from the post- to presynaptic terminal; the authors suggested that this functional asymmetry leads to cooperativity amongst the multiple converging auditory afferent axons thereby promoting the Mauthner escape response (*Rash et al., 2013*). Whether the configuration we find here of presynaptic Cx35.5 paired with post-synaptic Cx34.1 is sufficient to produce biased current flow across the channel is unclear; however Cx asymmetry may not act alone in biasing synaptic function. Cx function can be modified by differ-ences in the intracellular milieu such as Mg++ concentration (*Palacios-Prado et al., 2014b*) and dif-ferential phosphorylation (*Pereda, 2014*) or perhaps different pre- and postsynaptic protein interactions may bias function. The molecular asymmetry of Cxs we identify suggests that GJ-associ-ated proteins may also be asymmetrically distributed. Intriguingly, we previously found that the autism associated gene *neurobeachin*, which encodes a scaffolding protein thought to act within the trans-Golgi network, was required postsynaptically for electrical synaptogenesis (*Miller et al., 2015*). Future work is sure to extend our knowledge of molecular and functional asymmetry at electrical syn-apses where they are likely to be important for the development, function, and plasticity of neural circuits.

In our experiments we found that *gjd1a/cx34.1* and *gjd2a/cx35.5* are essential for the electrical synapses of the Mauthner circuit as well as many other synapses in zebrafish larvae. Previous freeze fracture immuno-EM work in goldfish has instead suggested that Cx35.1 and Cx34.7 may create the Aud/M synapses, yet we find that the genes encoding these two Cxs are not required in zebrafish (*Figures 3* and *5*). What accounts for this discrepancy? First, it is likely that the goldfish and zebrafish

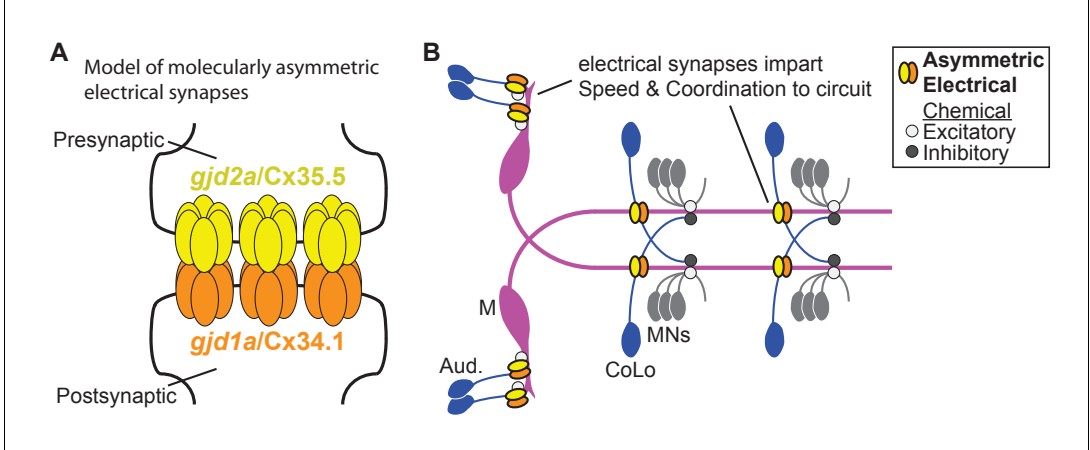

**Figure 9.** Schematics of the asymmetric functions of Cx proteins at zebrafish electrical synapses. (A) Model of an electrical synapse with an exclusive pre- and postsynaptic Cx making up the gap junction plaque. We provide genetic evidence of this asymmetry at Mauthner synapses but it is likely broadly used in the nervous system. Such asymmetry is likely to drive functional asymmetry at these synapses (see Discussion). (B) Model of asymmetric electrical synapses found in concert with chemical synapses of the Mauthner circuit. The electrical synapses contribute speed and coordination to circuit function. The speed is likely imparted via the Aud/M synapses in the hindbrain while the coordination is likely via the M/CoLo synapses in the spinal cord (see Discussion).

Aud/M synapses are constructed with the same Cxs, but this need not be the case and could account for differences. Moreover, the goldfish synapses were examined at adulthood while our work was in larval stages; it is conceivable that these electrical synapses could change their composition during development and maturation. Second, as others have shown (*Satou et al., 2009*; *Yao et al., 2014*), we found that antibodies generated against Cx34.7 and Cx35.1 and used in the goldfish studies (*Rash et al., 2013*) label zebrafish Aud/M and M/CoLo electrical synapses at larval stages from 3 to 7 dpf (see Materials and methods for antibody details). This, when put in the context of our genetic evidence for the requirement for *gjd1a/cx34.1* postsynaptically and *gjd2a/cx35.5* presynaptically, suggests that these previously used antibodies cross react with Cx34.1 and Cx35.5. Given the amino acid similarity of the two *gjd1* proteins and the two *gjd2* proteins (*Figure 2*), the most parsimonious scenario would be that the Cx34.7 antibody cross reacts with Cx34.1, while the Cx35.1 antibody cross reacts with Cx35.5. If correct, this would make for a consistent picture between the genetic evidence we present here and the goldfish immuno-EM data (*Rash et al., 2013*). Together, these data strongly support a model wherein the Aud/M synapses contain molecularly asymmetric gap junction channels. We also find that while Cx35.5 appears to be the sole presynaptic Cx at Aud/M synapses (*Figure 6K*), Cx34.1 may not act alone on the postsynaptic side of the synapse (*Figure 6H*). The nature of the 'unidentified' postsynaptic Cx is currently unknown but there are two main possibilities: (1) *gjd2a*/Cx35.5 may be used postsynaptically at these synapses or (2) an unidentified Cx is used postsynaptically to stabilize the observed Cx35.5 presynaptically. Whether the residual staining we observe in *gjd1a/cx34.1* mutants at Aud/M gap junctions represents further Cx complexity (i.e. multiple postsynaptic Cxs) or instead is the result of compensatory changes only found in the mutant will require future experiments to resolve. Our genetic and chimaeric evidence also support Cx molecular asymmetry at M/CoLo synapses in the spinal cord (*Figures 6* and *7*). This configuration is likely common in zebrafish given the broad colocalization of Cx34.1 and Cx35.5 and their frequent co-requirement for localization (*Figure 5*). Molecular asymmetries at electrical synapses are not confined to fish, but are found broadly throughout the nervous systems of *C. elegans* (*Starich et al., 2009*), medicinal leech (*Kandarian et al., 2012*), *D. melanogaster* (*Curtin et al., 1999*; *Phelan et al., 2008*), crab (*Shruti et al., 2014*), and there have been suggestions, though no definitive proof, that such may occur in mammals as well (*Cha et al., 2012*; *Haas et al., 2011*; *Zolnik and Connors, 2016*). If such does occur in mammalian systems it would have to rely on a different set of Cxs than in zebrafish as mammals appear to have lost the *gjd1/cx34* family of genes, leaving only the single, related *gjd2/cx36* gene (*Figure 2*). However, there are four other Cxs known to make electrical synapses in mammals (*Connors and Long, 2004*; *Söhl et al., 2005*; *Pereda, 2014*), and the extent of their use and the configurations of GJs they create is poorly understood. We expect that molecularly asymmetric electrical synapses are likely to be a part of all nervous systems given that they underlie important functional effects on the properties of neural circuit computation (*Phelan et al., 2008*; *Rash et al., 2013*).

How do the electrical synapses contribute to neural circuit computation and behavioral output? We found that the *cx* mutants are slower to respond to vibrational stimuli, but can still produce many fast escape responses consistent with Mauthner-induced activation (*Figure 8*). We prefer the model that the reduction in response speed is due to the loss of the electrical component of the Aud/M synapses (*Figure 9B*) given that these synapses are required for Mauthner activation and sufficient to create the fast escape response (*O'Malley et al., 1996*; *Tabor et al., 2014*). This idea is supported by electrophysiological recordings of Aud/M synapses in wildtype larval zebrafish at stages similar to those we analyzed where the electrical response occurs ~2 milliseconds faster than that of the chemical synaptic response (*Yao et al., 2014*). It is important to note that we cannot rule out the possibility that electrical synapses upstream of Mauthner are required for the observed defects, or that the loss of electrical synapses affects chemical synapse function within the Mauthner circuit. We also observed that a subset of escape responses in *cx* mutants display coordination defects. This is likely due to the loss of the M/CoLo synapses in the spinal cord as similar behavioral defects have been observed when CoLos are laser ablated or when strychnine is used to block inhibitory synapses (*Satou et al., 2009*; *Marsden and Granato, 2015*). These uncoordinated responses likely only occur in a subset of escapes given that the CoLos are productively engaged in the behavioral output only in cases where both Mauthner neurons fire in response to the stimulus (*Satou et al., 2009*). In wildtype when both Mauthners fire turning is unidirectional; this is achieved by the first-activated Mauthner's action potential exciting contralateral CoLos, which then recross

the spinal cord where they inhibit the motor neurons before the 'second-activated' Mauthner's action potential arrives – this ensures a unidirectional turn (see *Figure 9B* for circuit diagram) (*Fetcho, 1991*; *Satou et al., 2009*). When calcium activity of Mauthner is monitored in experimental paradigms similar to those we used it was found that both Mauthner neurons were activated at rates in line with the behavioral defects we observe (*Satou et al., 2009*; *Marsden and Granato, 2015*). Thus it is likely that the coordination defects we observe in mutants are due to the loss of the M/ CoLo synapses. Our results on Mauthner-induced behavior with and without neuronal gap junctions reveal a critical interplay between electrical and chemical synapses in creating the performance needed for the escape response. While we have focused mainly on the Mauthner circuit here, given the breadth of Cx34.1/Cx35.5-based electrical synapses observed, we expect to identify broad contributions of the electrical synapses to many behaviors in zebrafish. Indeed, loss of *gjd2/cx36* in mice results in defects in an ever increasing list of neurological functions including vision (*Güldenagel et al., 2001*; *Cowan et al., 2016*), smell (*Christie et al., 2005*), thalamic function (*Zolnik and Connors, 2016*), estrous cyclicity (*Campbell et al., 2011*), short-term spatial memory (*Allen et al., 2011*), motor performance (*Frisch et al., 2005*), as well as others. Future work in zebrafish and mouse will illuminate how electrical and chemical synapses collaborate in these complex, vertebrate nervous systems to produce appropriate behavior.

## Materials and methods

### Fish, lines, and maintenance

All animals were raised in an Institutional Animal Care and Use Committee (IACUC)-approved facility at the Fred Hutchinson Cancer Research Center (Study ID 50552, Submittal ID 7237, IRO # 1392). Zebrafish, *Danio rerio*, were bred and maintained as previously described (*Kimmel et al., 1995*). Rachel Garcia provided animal care and Dr. Rajesh K. Uthamanthil, DVM, provided veterinary care. *gjd1a*$^{fh360(dis3)}$ was isolated from an early-pressure, gynogenetic diploid screen (*Walker et al., 2009*) using ENU as a mutagen and animals were outcrossed from an *AB background into a Tuebingen background for mapping. *gjd1a*$^{fh436}$, *gjd1b*$^{fh435}$, *gjd2a*$^{fh437}$, and *gjd2b*$^{fh454}$ were generated using TALENs (*Sanjana et al., 2012*) targeting the first exon. *gjd2b*$^{fh329}$ was generated via TILLING (*Kettleborough et al., 2011*). See *Figure 2* and source data for information on all alleles. Stable zebrafish lines carrying each mutation were Sanger sequenced to verify mutations. All were maintained in the *M/CoLo:*GFP (*Et*(*Tol-056:GFP*)) background (*Satou et al., 2009*), which is a *AB/Tu background.

### RNA-seq-based mutant mapping

Larvae in the F3 generation were collected at three dpf from crosses of known *gjd1a*$^{fh360(dis3)}$ heterozygous animals, were anesthetized with MESAB (Sigma, A5040), and the posterior portion was removed and fixed for phenotypic analysis via immunohistochemistry (see below) while the anterior portion was placed in Trizol (Life Technologies, 15596–026), homogenized, and frozen to preserve RNA for sequencing. After phenotypic identification mutant (-/-) and wildtype sibling (+/+ and +/-) RNA was pooled separately from 108 embryos each. From each pool total RNA was extracted and cDNA libraries were created using standard Illumina TruSeq protocols. Each library was individually barcoded allowing for identification after multiplexed sequencing on an Illumina HiSeq 2000 machine. In brief, mapping was performed by identifying high quality 'mapping' single nucleotide polymorphisms (SNPs) in the wildtype pool and assessing these positions in the mutant pool for their frequency. The average allele frequency in mutants, using a sliding-window of 50-neighboring loci, was plotted across the genome and linkage was identified as the region of highest average frequency. Within the linked region candidate mutations causing nonsense or missense changes, or those affecting gene expression levels, were identified as previously described (*Miller et al., 2013*). Details can be found at www.RNAmapper.org.

### Cloning and characterization of connexin genes

We used the *gjd1a/cx34.1* sequences to search the Zv10 genome of zebrafish for other genes with similar sequence. From these searches we identified four loci, *gjd1a/cx34.1* (chr5), *gjd1b/cx34.7* (chr15), *gjd2a/cx35.5* (chr17), and *gjd2b/cx35.1* (chr20), as the only locations with significant

homology. All four are two-exon genes, however the gene encoding Cx35.5 is annotated on the zebrafish Zv10 genome as having the first exon downstream and in the opposite orientation of the second exon. This arrangement is most likely due to an inappropriately assembled region of the genome for three reasons: (1) we cloned a full-length transcript from a cDNA library using primers to the 5'- and 3'-UTRs, (2) the gene results in a protein product, (3) the gene results in a protein detectable by an antibody made against it (see results for points 2 and 3).

## Antibody generation and validation

Antibodies were generated at the Fred Hutchinson Antibody Technology Facility (https://sharedre-sources.fredhutch.org/core-facilities/antibody-technology). We used the variable regions of the intra-cellular loop from *gjd1a*/Cx34.1 and *gjd2a*/Cx35.5 (*Figure 2A,B*) to generate peptide antibodies to each protein and first tested the specificity of antibodies on each Cx protein individually expressed in HeLa cells. HeLa CCL-2 cells were obtained as an authenticated and mycoplasma free line from ATCC (ATCC Number: CCL-2) and the original cells and early passages of the cells were frozen back and maintained in liquid nitrogen. Cells were thawed and utilized under low passage conditions with all experiments done on lines split fewer than 30 times. HeLa cells were used as previous work established them as a standard method for Cx protein expression and antibody validation (*Rash et al., 2013*). We found several antibodies against each Cx that specifically recognized only the intended target Cx and not the others and screened each of these using whole-mount immunohistochemistry of 5 dpf zebrafish larvae. We found that these antibodies generated staining patterns highlighting the same structures in the midbrain, hindbrain, and spinal cord as staining with the antibody against human Cx36 used above (*Figure 5* and associated source data). The human anti-Cx36 antibody was also found to recognize all four zebrafish Cx36-related proteins (*Figure 5* and associated source data). We did not detect staining in areas anterior of the midbrain, but we did not perform the sectioning required to recognize electrical synapses in the retina and other anterior regions (*Li et al., 2009*). In *gjd1a/cx34.1* mutants we found that staining for Cx34.1 protein was lost while in the *gjd2a/cx35.5* mutants we found that Cx35.5 protein staining was lost supporting both the specificity of the antibodies as well as the deleterious nature of the mutations (*Figure 5*).

## Immunohistochemistry

Anesthetized embryos from 2 to 14 dpf were fixed in either 2% trichloroacetic (TCA) acid for 3 hr or 4% paraformaldehyde (PFA) for 1 hr. Fixed tissue was then washed in PBS + 0.5% TritonX100, followed by standard blocking and antibody incubations. Tissue was cleared step-wise in a 25%, 50%, 75% glycerol series and was dissected and mounted for imaging. Primary antibodies used were: chicken anti-GFP (abcam, ab13970, 1:250), rabbit anti-human-Cx36 (Invitrogen, 36–4600, 1:200), rabbit anti-Cx34.7 against the intracellular loop peptide of the perch protein (JOB, 1:20)(*O'Brien et al., 2004*), mouse anti-Cx35.1 against recombinant perch protein (EMD/Millipore, MAB3045, 1:200) (*O'Brien et al., 2004*), rabbit anti-Cx34.1 and mouse anti-Cx35.5 (both generated in this study, 1:100), mouse anti-RMO44 (Life Technologies, 13–0500, 1:100). All secondary antibodies were raised in goat (Life Technologies, conjugated with Alexa-405,–488, −555,–594, or −633 fluorophores, 1:250). Neurobiotin and Biotin Dextran were detected using fluorescently-tagged streptavidin (Life Technologies, conjugated with Alexa-633 fluorophores, 1:500).

## Neurobiotin retrograde labeling

Anesthetized 5 dpf embryos were mounted in 1% agar and a caudal transsection through the dorsal half of the embryo was made with an insect pin at somite 20–25. A second insect pin loaded with 5% neurobiotin (Nb) solution was quickly applied to the incision. Animals were unmounted from the agar and allowed to rest for 3 hr while anesthetized to allow neurobiotin to pass from Mauthner into the CoLos. Animals were then fixed in 4% PFA for 2 hr and processed for immunohistochemistry. CoLo axons project posteriorly for a maximum of two segments; therefore measurements of Nb in CoLo were analyzed at least three segments away from the lesion site.

## Cell transplantation

Cell transplantation was done using standard techniques at the blastula stage (*Kemp et al., 2009*). For 'mutant into wildtype' experiments examining the spinal cord M/CoLo synapses and assessing

the postsynaptic Mauthner of Aud/M synapses, animals heterozygous for a *cx* mutation and carrying the *M/CoLo:GFP* transgene were incrossed while hosts were non-transgenic AB; for 'wildtype into mutant', *M/CoLo:GFP* animals were incrossed while hosts were an incross of animals heterozygous for a *cx* mutation that were not transgenic. For assessing the presynaptic auditory afferent contribution to the Aud/M synapses embryos were injected with biotin-dextran at the 1 cell stage and later transplanted from appropriate genotypes. Approximately 20 cells were deposited 1–5 cell diameters away from the margin at the sphere/dome stage (4–4.5 hpf) with a single embryo donating to 3–5 hosts. Embryos were allowed to grow until 5 dpf at which point they were processed for immunostaining. Donor (mutant to wildtype) or host (wildtype to mutant) embryos were genotyped.

## Confocal imaging and analysis

All images were collected on a Zeiss LSM 700 confocal microscope using 405, 488, 555, and 639 laser lines, with each line's data being collected sequentially using standard pre-programmed filters for the appropriate Alexa dyes. All Z-stacks used 1 uM steps. Images were processed and analyzed using Fiji (*Schindelin et al., 2012*) software. Within each experiment all animals were stained together with the same antibody mix, processed at the same time, and all confocal settings (gain, offset, objective, zoom) were identical. For quantitating fluorescent intensity, synapses were defined as the region of contact between the neurons of interest (Aud/M or M/CoLo). A standard region of interest (ROI) surrounding each synapse was drawn and the mean fluorescent intensity was measured across at least 4 uMs in the Z-direction with the highest value being recorded. For neurobiotin backfills fluorescent intensity was measured using a standard ROI encompassing the entire Mauthner or CoLo cell body. Statistics were computed using Prism software (GraphPad). Figure images were created using Fiji, Photoshop (Adobe) and Illustrator (Adobe). Colors for all figures were modified using the Fiji plugin *Figure 5D*.

## Behavioral imaging and analysis

Behavioral experiments were performed on six dpf larvae as previously described (*Cruciani and Mikalsen, 2007*; *Wolman et al., 2011*, *2015*). Briefly, larvae were placed in individual wells of a 16-well grid and exposed to 3 ms, 1000 Hz acoustico-vibrational stimuli delivered by an acoustic shaker at 26 dB. Stimulus intensities were calibrated with a PCB Piezotronics accelerometer (model #355B04). All animals received 10 stimuli separated by 20 s, which is sufficient to ensure that no habituation occurs over the course of the experiment (*Wolman et al., 2011*). Movements were captured using a high-speed camera (RedLake MotionPro) at 1000 frames/sec, and movies were analyzed with the FLOTE software package as previously described (*Burgess and Granato, 2007*). Short latency C-bend startle responses (SLCs) driven by Mauthner were measured and defined by the previously established kinematic parameters of initiation latency, turning angle, turn duration, and maximum angular velocity (*Burgess and Granato, 2007*). After testing, larvae were transferred to 96-well plates and processed for subsequent genotyping.

## Acknowledgements

We thank Rachel Garcia for superb animal care, Kathryn Helde for help with the forward genetic screen, Lila Solnica-Krezel for ENU-mutagenized male zebrafish, Shin-Ichi Higashijima for the *M/CoLo:GFP* line, and the Fred Hutchinson Cancer Research Center's Genomic Resource Center, particularly Jeff Delrow, Andy Marty, Alyssa Dawson, and Ryan Basom, for sequencing library preparation, sequencing, and help in data processing. Funding was provided by the National Institute of Health, F32NS074839 and K99/R00NS085035 to ACM, R01MH109498 to MG, R01EY012857 to JO, and R01HD076585 and R21NS076950 to CBM.

## Additional information

### Funding

| Funder | Grant reference number | Author |
| --- | --- | --- |
| National Institute of Neurological Disorders and Stroke | F32NS074839 | Adam C Miller |

| National Institute of Mental Health | R01MH109498 | Michael Granato |
| National Eye Institute | R01EY012857 | John O'Brien |
| Eunice Kennedy Shriver National Institute of Child Health and Human Development | R01HD076585 | Cecilia B Moens |
| National Institute of Neurological Disorders and Stroke | R21NS076950 | Cecilia B Moens |
| National Institute of Neurological Disorders and Stroke | K99/R00NS085035 | Adam C Miller |

The funders had no role in study design, data collection and interpretation, or the decision to submit the work for publication.

## Author contributions

ACM, Conceptualization, Resources, Data curation, Formal analysis, Supervision, Funding acquisition, Validation, Investigation, Visualization, Methodology, Writing—original draft, Project administration, Writing—review and editing; ACW, Formal analysis, Validation, Investigation, Visualization, Methodology, Writing—original draft, Writing—review and editing; ANS, Investigation, Methodology; KCM, Conceptualization, Resources, Investigation, Methodology, Writing—review and editing; MG, Funding acquisition, Writing—review and editing; JO, Formal analysis, Funding acquisition, Investigation, Methodology, Writing—review and editing; CBM, Funding acquisition, Project administration, Writing—review and editing

## Author ORCIDs

Adam C Miller, http://orcid.org/0000-0001-7519-3677
Alex C Whitebirch, http://orcid.org/0000-0002-9178-1636
John O'Brien, http://orcid.org/0000-0002-0270-3442

## Ethics

Animal experimentation: All animals were raised in an Institutional Animal Care and Use Committee (IACUC)-approved facility at the Fred Hutchinson Cancer Research Center (Study ID 50552, Submittal ID 7237, IRO # 1392).

# Additional files

## Major datasets

The following dataset was generated:

| Author(s) | Year | Dataset title | Dataset URL | Database, license, and accessibility information |
| --- | --- | --- | --- | --- |
| Miller AC | 2013 | Dis2 RNA-seq wildtype and mutant | https://www.ncbi.nlm.nih.gov/sra/?term=PRJNA172016 | Publicly available at the NCBI Sequence Read Archive (accession no: PRJNA172016) |

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
