## [Decision Letter]

Thank you for submitting your article "A genetic basis for molecular asymmetry at vertebrate electrical synapses" for consideration by *eLife*. Your article has been reviewed by two peer reviewers, one of whom, Yang Dan, (Reviewer #1), is a member of our Board of Reviewing Editors and the evaluation has been overseen by Eve Marder as the Senior Editor.

The reviewers have discussed the reviews with one another and the Reviewing Editor has drafted this decision to help you prepare a revised submission.

Summary:

The authors show that electrical synapses used at many points along a motor circuit of the zebrafish larvae are asymmetric, comprised of connexins of two specific subtypes, polarized in a particular direction, and that this heterotypic arrangement is critical for normal development, function, and behavior. Specifically, the authors identified a mutation, dis3, that caused complete loss of electrical synapses in the circuit of interest (M/CoLo) without apparent changes to the rest of the circuit morphology or development. With genetic analyses they described the familial relationships of four neuronal connexins, Cx34.1, Cx34.7, Cx35.1, and Cx35.5, each of them homologues of mammalian Cx36. They found that dis3 is caused by mutations of the Cx34.1 gene. Genetic deletion of Cx34.1 or Cx35.5 led to complete loss of Cx36-like gap junctions and neurobiotin (NB)-coupling in the M/CoLo circuit. The authors also generated new antibodies specific for Cx34.1 and Cx35.5. Using a chimera approach they further showed that successful development of electrical synapses in the M/CoLo system depends on PREsynaptic expression of Cx35.5 and POSTsynaptic expression of Cx34.1. Testing of an M/CoLo-mediated behavior showed that Cx35.5/34.1 electrical synapses are necessary for full function.

Both reviewers agree that the study is impressive for its comprehensive, careful approach. The writing is exceptionally clear, and each Results section includes a welcome dose of background and rationale. While the specific findings are likely to hold only for teleost fish, the more general principles about electrical synapse asymmetry may turn out to be relevant to mammals and other classes.

Essential revisions:

1) Do mutants really have normal structure and function, apart from the loss of many electrical synapses and their functions? The authors state: "dis3 mutants display no gross morphological defects in nervous system morphology and neuronal number, have no defects in general body plan development, have normal developmental timing, are homozygous viable, and crosses between homozygous mutant animals produce viable offspring. We conclude that the gene disrupted by the dis3 mutation has a broad but specific role in electrical synapse formation." However, quantitative data to support this contention are not provided. If the authors have them, they should be.

2) In the Discussion, the authors say, "…we have found that the antibodies used in the previous study label all zebrafish Mauthner circuit electrical synapses at larval stages (data not shown)." This is an important point since, a) it addresses the discrepancy between their conclusions and those of Rash et al., 2013) and others, and b) it would be very interesting for other investigators in this field to know. Please at least provide the names and sources of the antibodies in question, and more specific description of the data "not shown".

---

## [Author Response]

*Essential revisions:*

*1) Do mutants really have normal structure and function, apart from the loss of many electrical synapses and their functions? The authors state: "dis3 mutants display no gross morphological defects in nervous system morphology and neuronal number, have no defects in general body plan development, have normal developmental timing, are homozygous viable, and crosses between homozygous mutant animals produce viable offspring. We conclude that the gene disrupted by the dis3 mutation has a broad but specific role in electrical synapse formation." However, quantitative data to support this contention are not provided. If the authors have them, they should be.*

We agree with the reviewers’ point that, in essence, there could be defects throughout the nervous system that were not apparent based on our experiments. We have therefore chosen to focus on the Mauthner circuit for this section and have included quantifiable results that suggest that the cells of the circuit and their morphology is intact. We have left out other anecdotal references to gross morphology, developmental timing, etc., as we have no quantifiable data. These changes are found in the Results section.

*2) In the Discussion, the authors say, "…we have found that the antibodies used in the previous study label all zebrafish Mauthner circuit electrical synapses at larval stages (data not shown)." This is an important point since, a) it addresses the discrepancy between their conclusions and those of Rash et al., 2013) and others, and b) it would be very interesting for other investigators in this field to know. Please at least provide the names and sources of the antibodies in question, and more specific description of the data "not shown".*

We appreciate the comment and the reviewers’ goal to make this information accessible to those in the field. In this section we now specifically reference the appropriate papers that have previously used the antibodies in question and we state directly what we have observed with these antibodies. These changes can be found in the Discussion. We have included the details about each antibody in the Material and methods section.